# Endoscopic Submucosal Dissection of Superficial Colorectal Neoplasms at “Challenging Sites” Using a Double-Balloon Endoluminal Interventional Platform: A Single-Center Study

**DOI:** 10.3390/diagnostics13193154

**Published:** 2023-10-09

**Authors:** Gianluca Andrisani, Francesco Maria Di Matteo

**Affiliations:** Digestive Endoscopy Unit, Fondazione Policlinico Universitario Campus Bio-Medico, Via Alvaro del Portillo 200, 00128 Rome, Italy

**Keywords:** EIP, colorectal polyps, ESD

## Abstract

Background: Colonic endoscopic submucosal dissection (ESD) at “challenging sites” such as the cecum, ascending colon, and colonic flexures could be difficult even for expert endoscopists due to poor endoscope stability/maneuverability, steep angles, and thinner wall thickness. A double-balloon endoluminal intervention platform (EIP) has been introduced in the market to fasten and facilitate ESD, particularly when located at difficult sites. Here, we report our initial experience with an EIP comparing the outcomes of an EIP versus standard ESD (S-ESD) at “challenging sites”. Materials and methods: We retrospectively collected data on consecutive patients with colonic lesions located in the right colon and at flexures who underwent ESD in our tertiary referral center between March 2019 and May 2023. Endoscopic and clinical outcomes (technical success, en bloc resection rate, R0 resection rate, procedure time, time to reach the lesion, and adverse events) and 6-month follow-up outcomes were analyzed. Results: Overall, 139 consecutive patients with lesions located at these challenging sites were enrolled (EIP: 31 and S-ESD: 108). Demographic characteristics did not differ between groups. En bloc resection was achieved in 92.3% and 93.5% of patients, respectively, in the EIP and S-ESD groups. Both groups showed a comparable R0 resection rate (EIP vs. S-ESD: 92.3% vs. 97.2%). In patients undergoing EIP-assisted ESD, the total procedure time was shorter (96.1 [30.6] vs. 113.6 [42.3] minutes, *p* = 0.01), and the mean size of the resected lesions was smaller (46.2 ± 12.7 vs. 55.7 ± 17.6 mm, *p* = 0.003). The time to reach the lesion was significantly shorter in the EIP group (1.9 ± 0.3 vs. 8.2 ± 2.7 min, *p* ≤ 0.01). Procedure speed was comparable between groups (14.9 vs. 16.6 mm^2^/min, *p* = 0.29). Lower adverse events were observed in the EIP patients (3.8 vs. 10.2%, *p* = 0.31). Conclusions: EIP allows results that do not differ from S-ESD in the resection of colorectal superficial neoplasms localized in “challenging sites” in terms of efficacy and safety. EIP reduces the time to reach the lesions and may more safely facilitate endoscopic resection.

## 1. Introduction

Endoscopic submucosal dissection (ESD) is the preferred technique to resect superficial colorectal neoplasms as it increases the proportion of successful en bloc resections, thus allowing a more accurate histological assessment. The high rate of curative interventions and the low risk of local recurrence make ESD particularly appropriate for the management of these lesions [1,2,3]. Colonic ESD at “challenging sites” such as the cecum, ascending colon, and colonic flexures is more arduous than rectal, esophageal, and gastric ESD. Indeed, at these sites, the maneuverability and stability of the endoscope are limited, the colonic angles may be steep, and the muscle layer is thinner and therefore the perforation risk is higher. Differently from other GI locations, in colonic ESD, the withdrawal and reinsertion of the endoscope are time-consuming [4]. Limited endoscopic operability and previously manipulated polyps resulting in severe fibrosis are known significant independent predictors of perforation [5]. Traction methods have been developed to provide sufficient lift for the tissue flap to provide better visualization of the dissection plane, thus allowing safer dissection while avoiding muscular layer injury [6,7]. Several traction approaches have been utilized such as clip-and-snare methods, the external forceps method, and a double scope method [7,8,9,10]. A double-balloon endoluminal interventional platform (EIP) (DiLumen™, Lumendi LLC, Westport, CT, USA) was introduced to help aid in the stabilization of the endoscope during endoscopic interventions in the colon (Figure 1). Furthermore, the device provides the ability to create a therapeutic zone (TZ) with the possibility of dynamic traction generated through the use of suture loops [11,12,13,14,15,16].

In this study, we evaluated our initial experience with the platform comparing the outcomes of EIP with the standard cap-assisted ESD (S-ESD) at “challenging sites”.

## 2. Patients and Methods

An observational, retrospective, single-center study at our tertiary referral site was undertaken to analyze clinical and endoscopic outcomes at baseline and at 6-month follow-up in consecutive patients with superficial colorectal lesions located at “challenging sites” (cecum, ascending colon, and colonic flexures) undergoing ESD using the EIP or S-ESD between March 2019 and May 2023. All the enrolled patients gave written informed consent to participate in the study, and the data were anonymized according to current regulations. The study protocol was approved by the local independent ethics committee (Approval: Prot.:PAR 79.23 OSS ComEt-CBM). This research received no specific grant; it is a no-profit study. All procedures were performed in accordance with the Declaration of Helsinki [17].

Demographic and clinical characteristics of the study population, including lesion locations, are shown in Table 1 and Table 2.

### 2.1. Endoscopic Procedures

All resections were performed by experienced operators. Lesions were classified using the Paris classification and the classification for laterally spreading tumors (LST) [18,19].

### 2.2. Double-Balloon Endoluminal Intervention Platform (EIP)

The double-balloon endoluminal intervention platform (EIP, DiLumen™, Lumendi LLC (Westport, CT, USA)) is a non-sterile single-use commercially available endoscopic overtube. The DiLumen™ device includes a polyurethane sheath (103 cm, 130 cm) and two balloons that can be independently inflated or deflated. The Aft-Balloon (AB) is fixed behind the endoscope tip, and the Fore Ballon (FB) can be advanced or retracted beyond the endoscope tip. The “TZ” can be created by fully extending the FB and inflating both balloons by the inflation handle (Figure 2).

After the device is loaded over the colonoscope, both are advanced together to the lesion site with the balloons deflated. If needed, the AB can be inflated/deflated sequentially to assist with colon reduction in a sinuous or long colon during navigation, using a method similar to balloon-assisted small bowel enteroscopy [20], with the advantage of not using fluoroscopy.

Both balloons can be inflated up to a diameter of 6 cm and an internal pressure of 55 mmHg. The AB provides endoscope tip stability by gripping the colonic mucosa, and the adjustable FB provides the ability to flatten folds, reduce flexures, and provide tissue retraction. This device is also equipped with 2 pre-placed sutures (a long and a short suture) on the FB to allow for either an adjustable push or pull method of “dynamic retraction” of the lesion with the use of an endoscopic clip as dissection proceeds (Appendix A). In our experience, we used two different versions of DiLumen™. The substantial difference between the two versions is the hydrophilic coating. In the first version, it was necessary to use a water-based lubricant to allow the colonoscope to pass through the sleeve. In the present “EZ Glide version”, only water is needed to lubricate the inner sheath of the device (Figure 3).

Each patient first underwent a colonoscopy to identify and evaluate the lesion. The EIP was secured over the colonoscope, and both advanced through the anus after lubrication. In the case of EIP failure, the device was removed and S-ESD was performed.

### 2.3. Endoscopic Submucosal Dissection

Endoscopic submucosal dissections were performed by using Fujifilm colonoscopes 9 (Fujifilm, Tokyo, Japan) (EC-760R-VL/EC-760ZP-VL). CO_2_ gas was used in all cases. Two types of electrosurgical generators (VIO 300D or VIO 3, ERBE, Tübingen, Germany) were used. The knives were Flush-knife (Fujifilm, Tokyo, Japan) or HybridKnife (Erbe Elektromedizin, Tübingen, Germany). The technique adopted for the ESD (Figure 4) was chosen according to operator preference and included either the “conventional” [21], pocket [22], or tunnel technique [23]. Various traction methods to better expose the submucosal layer were applied [24]. Local injection was performed with either saline or glyceol with indigo carmine.

### 2.4. Histopathological Evaluation

All resected lesions were fixed in formalin and cut into 2 mm thick sections. Pathological evaluations were reported according to the Vienna classification [25].

The depth of invasion was classified according to the SM classification and defined as SM1 (<1000 μm from muscolaris mucosae) or SM2 (≥1000 μm from muscolaris mucosae).

The endoscopic resection was defined as “en bloc” if the lesion was resected in its entirety. If both lateral and vertical margins were free from neoplasia in histology, then the resection was defined as “R0”. Differently, if neoplasia was identified at the horizontal or lateral margins, it was defined as an “R1” resection. Finally, endoscopic resection was considered “curative” in the case of R0 resection with SM1 invasion, absence of lymphovascular invasion, low tumor budding, and G1-G2 differentiation [26].

### 2.5. Adverse Events and Follow-Up

Adverse events (AEs) were recorded and included mucosal damage, delayed bleeding, colonic perforation, and post-ESD electrocoagulation syndrome (PECS).

Late bleeding was defined as melena or rectal bleeding or >2 g/dL hemoglobin level drop requiring endoscopic hemostasis [27,28]. Perforation was considered significant as a complication if treatment with endoscopic clips or surgery was needed. PECS was diagnosed on the basis of the inflammatory response condition (fever, abdominal pain, and leukocytosis) post ESD, with no evidence of perforation by abdominal X-ray or CT scan.

A follow-up colonoscopy was planned at 6 months, according to current clinical practice. For all the enrolled patients, biopsies of the ESD scar were routinely taken.

### 2.6. Study Outcomes

The primary outcomes were efficacy and safety of ESD with EIP, considering technical success, defined as the achievement of the position to undergo the approach to the lesion in ESD with the EIP; en bloc resection rate, defined as the rate of en bloc resection of the lesion; R0 resection rate, defined as the percentage of patients with histologically negative lateral and deep margins. The secondary outcomes were time to reach the lesion after placing the EIP; procedure speed (mm^2^/min), calculated by dividing the area of the resected piece (mm^2^) over procedure time (minutes); procedure time, calculated for ESD from submucosal injection until complete dissection of the lesion; area of the resected piece; AEs.

## 3. Statistical Analysis

Baseline demographics and clinical characteristics of patients and lesions were reported as mean (±SD). Differences between groups were assessed using the chi-square test and Student’s *t*-test, as appropriate. Differences were considered statistically significant for a *p* value < 0.05.

## 4. Results

Between March 2019 and May 2023, 139 patients with colonic lesions located at “challenging sites” undergoing ESD were enrolled, including 108 S-ESD and 31 EIP cases. Demographics and clinical characteristics of the study population and of the colonic lesions are reported in Table 1. Among the 139 patients enrolled, 5 were excluded as the first version of the EIP was used and ESD was completed without EIP. The age and sex distributions are reported in Table 1 and were comparable between groups (Table 1). Overall, technical success was achieved in 96.1% of patients treated with the new version of the device (DiLumen EZ-Glide) (Table 2). In one patient, it was not possible to create the TZ and the device was only used to reach the ascending colon.

The lesion location in the EIP group included the ascending colon in 17, the cecum in 2, the hepatic flexure in 5, and the splenic flexure in 2 patients. No statistical difference in lesion location was observed between the EIP and S-ESD groups. En bloc resection was achieved in 92.3% and 93.5% of patients, respectively, in the EIP and S-ESD group and was statistically comparable. The R0 resection rate did not differ between groups, being 92.3% in EIP-treated patients vs. 97.2% in S-ESD patients, with S-ESD showing a higher rate of R0, which, however, did not reach statistical significance. In the EIP group, in one patient, the lateral margin of the resection specimen was not free of dysplasia (Rx), and the patient was therefore referred for endoscopic full-thickness resection at the follow-up colonoscopy. Lesions treated with ESD with EIP were significantly smaller than those treated with S-ESD (46.2 mm [12.7] vs. 55.7 mm [17.6], *p* = 0.003) and presented a significantly smaller surface area (1458.2 mm^2^ [824.5] vs. 1896.7 mm^2^ [1287.7], *p* = 0.03). In patients with EIP-assisted ESD, a significantly shorter total procedure time was observed (96.1 vs. 113.6 min; *p* = 0.01), while procedure speed was comparable between groups (14.9 vs. 16.6 mm^2^/min, *p* = 0.29). In the EIP group, the time to reach the lesion was significantly shorter (1.9 ± 0.3 vs. 8.2 ± 2.7 min, *p* ≤ 0.01).

Overall, in the EIP group, a lower AE rate was observed even though it was non-statistically significant (3.8 vs. 10.2%, *p* = 0.3). We recorded one case of perforation in the EIP group, while in the S-ESD group we found bleeding in two patients, four cases of perforations, and PECS in five patients; most PECS occurred in patients with large lesions in the cecum. All complications were managed conservatively.

When excluding the 5 surgically treated patients for advanced adenocarcinoma and the 11 patients lost to follow-up, 88% of the patients treated with ESD with EIP and 86% of the patients treated with S-ESD underwent a follow-up colonoscopy at 6 months.

Histologically, no difference was observed between groups, which showed a comparable occurrence of histological subtypes. Local recurrence was detected in 3.8% of the EIP group and 2.7% of the S-ESD group. Full outcomes are available in Table 3.

## 5. Discussion

In our early experience, we compared EIP-ESD and S-ESD for the treatment of lesions located at “challenging sites”, including the cecum, the ascending colon, and the colonic flexures. In these colonic sites, ESD can be difficult and is associated with a higher risk of adverse events even for expert endoscopists. Previous clinical studies have reported that tumor location in “challenging sites” is a risk factor for intestinal perforation during colorectal ESD, which also includes lesion dimensions, fibrosis, poor endoscopic operability, and the endoscopist’s experience [29,30,31]. The double-balloon platform was conceived to try to solve at least some of these issues in order to reduce perforation risk. The EIP serves as an anchor on the colon wall, reducing sigmoid looping and shortening the colon while providing optimized visualization and more stable access to the lesion. Furthermore, the creation of a TZ associated with the availability of traction to the lesion allows a better visualization of the cutting plane and a reduction in the insufflation of CO_2_ or, in the case of an underwater technique, of the volume of liquid. Our results showed high technical success with the new version of the device (DiLumen EZ-Glide) consistent with prior data reported in the literature (92–100%) [11,12,13,14,15,16]. This result is promising, and it could change the ESD technique, making this platform and consequently ESD accessible to everyone without the need for a particularly steep learning curve. In addition, the usefulness of EIP not only improves the movement of the endoscope during the ESD procedure but, by allowing dissection of the submucosa just above the muscle layer, it could increase the en bloc/R0 resection rate of young trainees achieving adequate pathological evaluation after ESD. This result is in contrast to our early experience with this platform for ESD as the endoscope remained fixed to the sheath, making it impossible to move and control the tip. The company changed the platform in late 2020, making the inside lining of the sheath hydrophilic, which reduced device preparation time and friction, making it easier to slide the endoscope within the sheath and improve control of the endoscope tip, which is essential for performing ESD. In our study population, an en bloc resection rate of 92.3% with the EIP was observed. In two patients during the traction phase with the EIP, we fragmented the lesion. This is probably a consequence of overextending the FB using the handle slider knob to provide traction to the mucosa. We recorded a high and comparable rate of R0 resection in both groups (92.3% vs. 97.2 for EIP and S-ESD, respectively). Of note, the presence of the two FB “pushrods” in the “therapeutic zone” can make it difficult to dissect the edges of the lesion, resulting in an RX of the lateral margin of the lesion. Therefore, it is necessary to complete the circumferential incision and part of the dissection before clipping the mucosal flap to the FB.

Long procedural time is a known downside of colorectal ESD and can represent a potential strength of EIP. Ismail et al. [12] did not report differences in terms of procedural time in the EIP vs. conventional ESD groups (mean ± SD: 81.9 ± 35.4 min S-ESD vs. 96.4 ± 42.2 min in ESD with EIP), despite the mean polyp size being comparable between groups (7.6 ± 6.0 cm^2^ vs. 6.2 ± 5.5 cm^2^, *p* = 0.2). Interestingly, in our study, the procedural time was shorter in patients undergoing ESD with EIP than in patients treated with S-ESD (*p* = 0.01). However, it should be acknowledged that in our study cohort, EIP was used for significantly smaller lesions when compared to S-ESD (46.2 mm vs. 55.7 mm, *p* = 0.003). The larger size of lesions treated with S-ESD could be due to the varying technical approaches. Indeed, the presence of the two pushrods in the “therapeutic zone” makes it difficult to dissect large lesions as the pushrods themselves could hinder the lift of the lesion.

An interesting result of our study is the time to reach the lesion, which was significantly shorter in the EIP group (1.9 ± 0.3 vs. 8.2 ± 2.7 min, *p* ≤ 0.01). This provides a great advantage for the operator, who must deal with the high risk of perforation due to lesions located at “challenging sites”. Direct and fast access to the resection site allows the operator to switch endoscopes and, more importantly, allows the timely management of any complications with available devices (over-the-scope clips or suturing devices). Differently from Ismail et al., we reported a significantly lower complication rate with the EIP (3.8 vs. 10.2%, *p* = 0.31). This could be explained by a reduced risk of perforation and thermal damage during dissection, following the countertraction applied on the submucosa by the double-balloon EIP associated with lifting after submucosal infiltration, thus better exposing the cutting plane. Another advantage of the EIP could be easier specimen retrieval, since the specimen is fixed at the tip of the device, making it unnecessary to use other devices, and leading to a possible reduction in terms of procedural costs. Limitations of this study include the relatively small study population, the retrospective design of the study, and the involvement of only one center. Indeed, the study was conducted in a referral center with expert endoscopists; thus, our results may not be applicable in all clinical settings and, in particular, in low-volume centers. Moreover, the overall costs of the EIP procedure were not calculated. Despite these limitations, we conclude that the double-balloon platform is safe and facilitates reaching the procedural site, representing a new and useful device in the ESD “toolbox”.

## Figures and Tables

**Figure 1 diagnostics-13-03154-f001:**
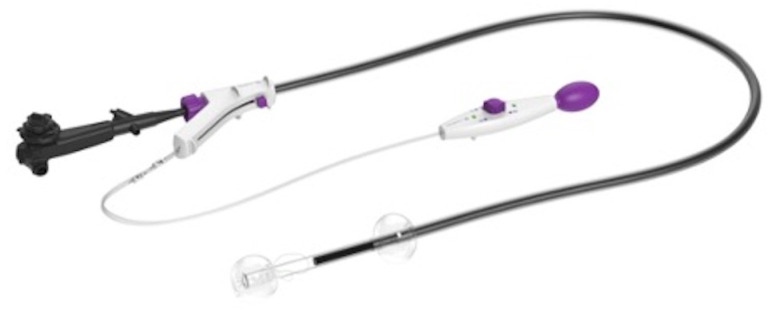
Double-balloon endoluminal intervention platform (EIP).

**Figure 2 diagnostics-13-03154-f002:**
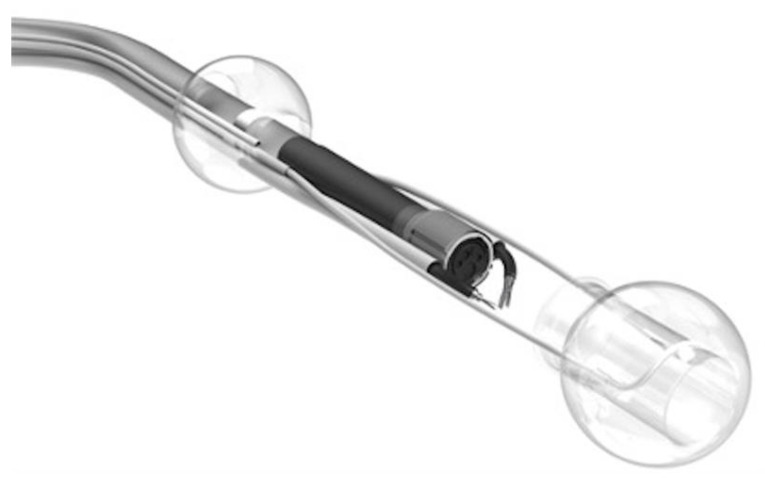
A therapeutic zone is created when the FB is extended beyond the endoscope tip and both balloons are inflated by the inflation handle.

**Figure 3 diagnostics-13-03154-f003:**
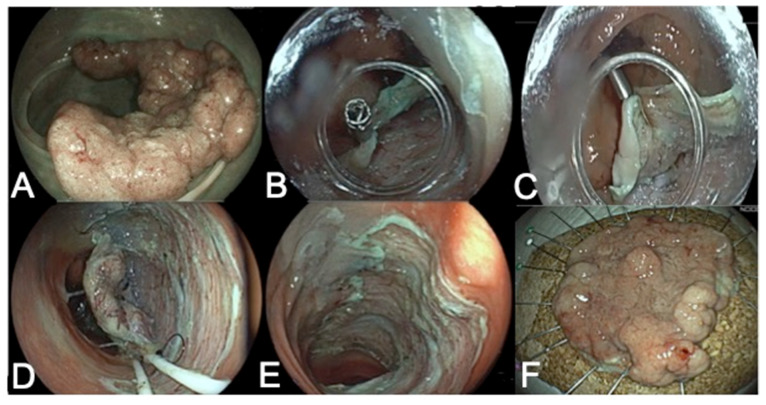
EIP-ESD procedure: (**A**) large laterally spreading tumor of splenic flexure; (**B**) the traction loops of the fore balloon are attached to the anal side of the lesion using a clip; (**C**) the fore balloon is extended slightly to create sufficient tension for rapid dissection; (**D**) dynamic retraction of the fore balloon; (**E**) en bloc resection site; (**F**) en bloc resection specimen.

**Figure 4 diagnostics-13-03154-f004:**
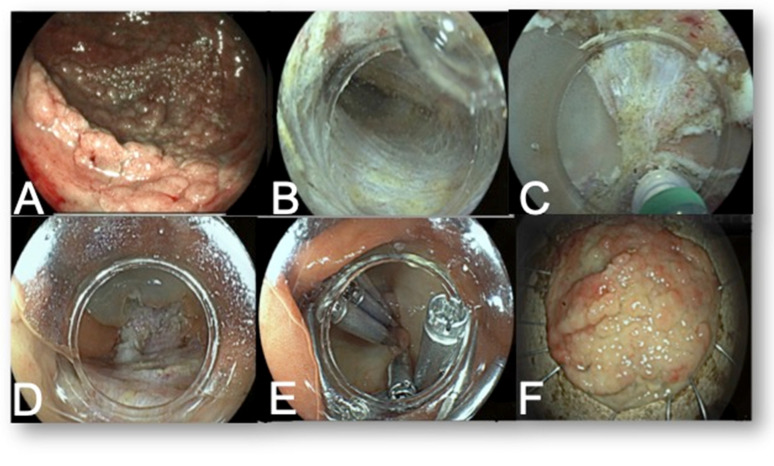
Colorectal ESD procedure. (**A**) Large laterally spreading tumor of cecum; (**B**) submucosal tunnel; (**C**) “final cut”; (**D**) en bloc resection site; (**E**) post-resection closure with clips; (**F**) en bloc resection specimen.

**Table 1 diagnostics-13-03154-t001:** Study patients’ characteristics.

	EIP	S-ESD	*p* Value
TOTAL	31	108	
Men, *n* (%)	18 (65.3)	59 (64.1)	0.9
Age, average ± SD	68.6 (8.4)	68.2 (10.8)	0.83
Anatomical site of the lesions, *n* (%)			
Splenic flexure	3 (9.6)	7 (6.5)	0.55
Hepatic flexure	6 (19.3)	13 (12)	0.29
Ascending colon	20 (64.5)	66 (61.1)	0.86
Cecum	2 (6.4)	22 (20.4)	0.02
The Paris classification, *n* (%)			
Is	2 (6.4)	7 (6.4)	1
0-IIa	2 (6.4)	4 (3.7)	0.51
0-IIb			
0-IIc			
LST-GM	13 (42)	46 (42.6)	0.95
LST-GU	2 (6.4)	12 (11.1)	0.44
LST-NG-F	6 (19.4)	23 (21.3)	0.81
LST-NG-PD	6 (19.4)	16 (14.9)	0.54

**Table 2 diagnostics-13-03154-t002:** Characteristics of subjects who underwent successful EIP-ESD.

	EIP	S-ESD	*p* Value
TOTAL	26	108	
Men, *n* (%)	17 (65.3)	59 (64.1)	0.9
Age, average ± SD	67.5 (7.9)	67.8 (10)	0.87
Anatomical site of the lesions, *n* (%)			
Splenic flexure	2 (7.7)	7 (6.5)	0.82
Hepatic flexure	5 (19.2)	13 (12)	0.33
Ascending colon	17 (65.4)	66 (61.1)	0.94
Cecum	2 (7.7)	22 (20.4)	0.13
The Paris classification, *n* (%)			
Is	2 (7.7)	7 (6.4)	081
0-IIa	2 (7.7)	4 (3.7)	0.37
0-IIb			
0-IIc			
LST-GM	10 (38.5)	46 (42.6)	0.7
LST-GU	2 (7.7)	12 (11.1)	0.61
LST-NG-F	4 (15.4)	23 (21.3)	0.5
LST-NG-PD	6 (23)	16 (14.9)	0.31

**Table 3 diagnostics-13-03154-t003:** Study outcomes.

	EIP	S-ESD	*p* Value
	26	108	
En bloc resection, *n* (%)	24 (92.3)	101 (93.5)	0.8
R0 resection, *n* (%)	24 (92.3)	105 (97.2)	0.2
Specimen diameter, average (SD), mm	46.2 (12.7)	55.7 (17.6)	0.003
Time of procedure, average ± SD, min	96.1 (30.6)	113.6 (42.3)	0.01
Time to reach the lesion, average ± SD, min	1.9 (0.3)	8.2 (2.7)	<0.01
Speed of procedure (mm^2^/min)	14.9 (6.8)	16.6 (8.3)	0.29
Area (mm^2^), average (SD)	1458.2 (824.5)	1896.7 (1287.7)	0.03
Histology, *n* (%)			
LDG adenoma (Low Grade Displasia)	2 (7.7)	3 (2.8)	0.23
HDG adenoma (High Grade Displasia)	12 (46.1)	60 (55.5)	0.38
Intramucosal Carcinoma	7 (27)	31 (28.8)	0.85
Sm1 Carcinoma	2 (7.7)	11 (10.1)	0.23
Sm2 Carcinoma	2 (7.7)	1 (0.9)	0.03
Sm3 Carcinoma	1 (3.8)	2 (1.9)	0.56
Complications *n* (%)	1 (3.8)	11 (10.2)	0.31
Perforation	0	4	
Late Bleeding	0	2	
PECS	0	5	
Follow-up			
Recurrence, *n* (%)	1 (3.8)	3 (2.7)	0.76

## Data Availability

Data available in a publicly accessible repository.

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
