# Peer review of "Endoscopic Submucosal Dissection of Superficial Colorectal Neoplasms at “Challenging Sites” Using a Double-Balloon Endoluminal Interventional Platform: A Single-Center Study"

_diagnostics, 2023, doi:10.3390/diagnostics13193154_

Round 1

Reviewer 1 Report

In this retrospective study, authors evaluated their initial experience of double balloon endoluminal interventional platform comparing the outcomes with the standard cap-assisted ESD at challenging sites such as colonic flexure parts. Although this study was conducted as an observational, retrospective, single center study, the main research findings of this paper will be important for performing safe and efficient ESD. I have no practical criticisms in terms of methods, results and interpretation.

Author Response

Thank you so much for taking the time to review this manuscript and for your positive comments

Reviewer 2 Report

Dear authors

your paper concerns a very interesting and debated issue. The technical improvement presented in your study has to be further explained and debated.

The results are very interesting but the number of patients treated by means of EIP is still too low.

Can you detail in how many case did you try to use EIP without success?

Discuss these issue in the results and discussion

Author Response

Thank you very much for taking the time to review this manuscript.

1.Certainly, the relatively small study population does not allow us to make laudatory statements. We reiterated in the results:" This result is promising and it could change the ESD technique making this platform and consequently ESD accessible to everyone without the need for a particularly long learning curve. In addition, the usefulness of EIP not only improves the movement of the endoscope during the ESD procedure, but by allowing dissection of the submucosa just above the muscle layer, it could increase the en bloc/R0 resection rate of young trainees achieving adequate pathological evaluation after ESD"

  1. Can you detail in how many case did you try to use EIP without success?

The question is appropriate. And we reported in the text:"In our experience we used two different versions of DiLumen™. The substantial difference between the two versions is the hydrophilic coating. In the first version it was necessary to use a water-based lubricant to allow the colonoscope to pass through the sleeve. In the present “EZ Glide version”, only water is needed to lubricate the inner sheath of the device 5 were excluded as the first version of the EIP was used and ESD completed without EIP. We have reported in table 1 the characteristics of all the patients in which we used the EIP and in table 2 only of the patients in which we used the new version.”